# Molecular Targets for Intracranial Aneurysm Treatment

**DOI:** 10.3390/ijms262010053

**Published:** 2025-10-15

**Authors:** Hunter Hutchinson, Rogina Rezk, Mariam Farag, Abanob Hanna, Brandon Lucke-Wold

**Affiliations:** 1College of Medicine, University of Florida, Gainesville, FL 32610, USA; hutchinsonhunter@ufl.edu (H.H.); roginarezk@ufl.edu (R.R.); mariamfarag@ufl.edu (M.F.); ahanna3@ufl.edu (A.H.); 2Department of Neurosurgery, University of Florida, Gainesville, FL 32608, USA

**Keywords:** intracranial aneurysm, inflammation, hemodynamics, endovascular therapy, matrix metalloproteinases, nuclear factor κB, interleukin-6, monocyte chemoattractant protein-1, tissue inhibitors of matrix metalloproteinases, nitric oxide synthase

## Abstract

Intracranial aneurysms (IAs) are a common cerebrovascular pathology with deadly potential. Neurointerventionalists commonly treat IAs with endovascular coiling, minimizing procedural risk at the cost of an increased recurrence rate. New therapies for reducing the rate of coiled and uncoiled IA growth and rupture would help reduce the morbidity and mortality patients experience when IAs rupture. Hemodynamic shear stress drives IA formation through molecular mechanisms, generating damage-associated molecular proteins (DAMPs), which lead to inflammation and extracellular matrix remodeling. Nuclear factor κB (NF-κB) and interleukin-6 (IL-6) maintain an inflammatory environment in IA walls, generating immune-cell chemotactic proteins, such as monocyte chemoattractant protein-1 (MCP-1) and IL-8. These molecules play a complex role in IAs, being important for IA formation and IA healing. Vascular smooth muscle cells and infiltrated immune cells secrete matrix metalloproteinases (MMPs), which initiate extracellular matrix remodeling. Tissue inhibitors of matrix metalloproteinases (TIMPs) balance this remodeling. The increased MMP to TIMP ratio is characteristic of IA progression, making these molecules important targets for IA therapies. Endothelial dysfunction generates nitric oxide and other reactive oxygen species, which exacerbate inflammation and cell death in IA walls. A better understanding of molecular mechanisms underlying IA formation, progression, and rupture will allow researchers to develop molecular IA therapies.

## 1. Introduction

Intracranial aneurysms (IAs) are a common cerebrovascular pathology with deadly potential. Magnetic resonance imaging (MRI) scans detect IAs in 2.3% of all patients [1]. Epidemiologically, non-modifiable risk factors for IA include a family history of cerebrovascular disease [2], female sex [2], connective tissue diseases [3], autosomal dominant polycystic kidney disease [4], and certain genetic polymorphisms [5]. Modifiable risk factors for IA include hypertension and smoking [5]. The majority of IAs are asymptomatic [6]. There is an overall 1.9% yearly risk of aneurysm rupture, leading to subarachnoid hemorrhage (SAH) [7]. Aneurysmal SAH has a high mortality rate of almost 50% and 22.8% of patients who survive aneurysmal SAH have severe neurological disabilities 5 years after IA rupture [7,8]. This risk increases with cocaine use and a family history of cerebrovascular disease [9,10]. IAs are abundant and frequently treated by neurointerventionalists because of their substantial risk.

The shared decision-making between patients and their physicians about how to treat IAs balances the mortality and morbidity of SAH with the cost and risk of surgical intervention. Initial monitoring for IAs less than 7 mm is a dual-subtraction cerebral angiogram every 6 to 12 months, with a 1-to-2-year follow-up after determination of IA stability [3,11]. Neurointerventionalists treat larger IAs with endovascular coiling or open-surgical clipping [12]. Endovascular coiling is the preferred method for IA treatment because it leads to fewer neurological deficits and shorter hospital stays [13]. However, coiled IAs still have a 10% recurrence rate within the first year and continue to recur at longer time periods [14]. Alternatively, clipped IAs have a yearly recurrence rate of only 0.7% [15]. Endovascular coiling is the most common treatment for IAs, minimizing procedural risk at the cost of increased recurrence rate.

Because of the prevalence of endovascular coiling despite its increased risk of recurrence, new therapies for reducing the rate of coiled and uncoiled IA growth and rupture are a critical area of research. An ideal therapy targets the pathophysiological components of IAs: inflammatory response, extracellular matrix (ECM) remodeling, and oxidative stress. The purpose of this narrative literature review is to overview the molecular components of IA formation and highlight target molecules for IA therapeutics in each category.

## 2. Inflammatory Pathways

Hemodynamic stress drives IA formation through molecular mechanisms [16]. Blood flow through cerebral arteries creates two types of stress on the endoluminal wall: tangential pressure and shear stress [16]. The initial insult leading to IA formation is high shear stress, exacerbated by hypertension, which disrupts the endothelial cell layer [16,17]. For this reason, IAs tend to occur at bifurcations or sharp curves in the cerebral vascular system [18]. Endothelial disruption by shear stress causes a phenotypic change in endothelial cells, leading to the release of inflammatory molecules and upregulation of molecules that modulate vascular tone, such as nitric oxide (NO) [19,20,21]. The tortuous, multifarious nature of cerebral vasculature creates a unique profile of hemodynamic stress, leading to endothelial release of targetable IA-forming molecules.

Hemodynamic stress draws numerous inflammatory cells to sites of IA formation [20]. Macrophage infiltration is characteristic of unruptured IAs, an attempt to repair damaged IA tissue [22]. This initiates an inflammatory cascade mediated by nuclear factor κB (NF-κB) that remodels and weakens the vascular wall [23]. Continued hydrodynamic stress causes vascular smooth muscle cells (VSMCs) to shift the phenotype from contractile to synthetic [24]. Synthetic VSMCs also initiate inflammation via NF-κB while secreting additional cytokines and matrix metalloproteinases (MMPs) that lead to ECM remodeling and IA growth [25,26]. Molecular approaches to attenuate inflammatory-cell response in the tunica media and tunica adventitia could alleviate vessel weakening and IA formation.

Toll-like receptor-4 (TLR4) activation of NF-κB mediates inflammation in IA formation [27]. Hemodynamic stress physically damages endothelial cells, which release damage-associated molecular patterns (DAMPs) that activate TLR4 signaling, initiating the synthesis of interleukin-8 IL-8, chemokine receptor ligand 1 (CXCL1), stromal-derived factor-1α (SDF-1α), and monocyte chemoattractant protein-1 (MCP-1) within endothelial cells [28,29,30,31]. Blood in human IAs contains higher concentrations of inflammatory markers than plasma from the same patients, attracting inflammatory cells to the IA site [32]. High concentrations of these molecules also exist in IA walls, leading to neutrophil infiltration [30]. Overall, NF-κB is a central regulator of inflammatory pathways triggered by vascular injury with important molecular targets for IA therapies.

A positive feedback loop maintains chronic inflammation within the aneurysm wall of IAs. NF-κB activation increases IL-6 transcription, augmenting NF-κB signaling [33]. This pro-inflammatory environment leads to the development of aneurysms and the advancement of vascular injury [33,34]. In addition, tumor necrosis factor-α (TNF-α) is a pro-inflammatory cytokine induced by tissue injury. It primarily signals through its receptor, tumor necrosis factor receptor-1 (TNFR1), leading to activation of the NF-κB pathway via a mechanism distinct from DAMP-induced TLR4 signaling [35]. It is important to note that, in addition to its pro-inflammatory effects, TNF-α also induces phenotypic modulation of VSMCs, promoting a remodeling state characterized by the upregulation of MMPs [36,37].

Balancing MCP-1 expression at distinct phases of IA formation is important for IA treatment. MCP-1 is a chemokine that functions by attracting monocytes and macrophages to areas of vascular damage, leading to further inflammation and weakening of the vessel wall in the initial stages of IAs [38]. Preventing initial MCP-1 expression on the precipice of IA formation is a futile strategy, as clinicians do not screen for IAs to detect early-stage formation [1]. Instead, attempts at molecular therapies in murine models of IAs attempt to increase the therapeutic occlusion caused by IA wall hyperplasia by endovascular coiling with MCP-1-coated coils [39]. In a murine model, MCP-1-coated coils significantly increased IA lumen ingrowth with significant macrophage, VSMC, endothelial cell, and fibroblast migration, histologically resembling successfully occluded human IAs [39,40]. MCP-1 attracts inflammatory cells that initiate IA formation and IA ingrowth, making the delivery or upregulation of MCP-1 to IAs a promising molecular therapy.

IL-8 is a neutrophil chemotactic molecule that is important in promulgating IA wall inflammation and damage. The ratio of IL-8 in the cerebrospinal fluid versus IL-8 in plasma is higher in patients with unruptured IAs [41]. However, there is insufficient evidence that IL-8 from IAs can cross the blood–brain barrier, a key limitation to interpreting the role of IL-8 in IA formation. Reminiscent of macrophage inflammation caused by MCP-1, neutrophil invasion of the aneurysm wall induced by IL-8 exacerbates inflammation and causes ECM remodeling [42]. Animal models have implicated neutrophils in IA rupture by the secretion of MMP9 [43]. Neutrophil extracellular traps (NETs) formed by extravasated neutrophils exist in ruptured IAs, interestingly less prevalent in estrogen-depleted mice [44]. The significant role of IL-8 in IA formation and rupture makes it a molecule of interest for IA therapies, whether directly or via estrogen modulation.

Given the important roles of the inflammatory markers, it is noteworthy that there are anti-inflammatory strategies targeting these pathways, such as BAY 11-7082 [45]. BAY 11-7082 is a well-characterized NF-κB inhibitor that irreversibly blocks IκBα phosphorylation, thereby preventing p65/p50 nuclear translocation and subsequent transcription of pro-inflammatory genes [45]. In addition, it suppresses the activation of activator protein-1, interferon regulator factor 3, and signal transducer and activator of transcription-1 through inhibition of extracellular signal-regulated kinase, p38, TANK-binding kinase-1, and Janus kinase-2, demonstrating a broad anti-inflammatory profile that extends beyond the established IKK/NF-κB pathway inhibition [45]. Although BAY 11-7082 is a known inhibitor of NF-κB, there have been no direct clinical studies conducted in patients with IAs, but it is a tool to track the inflammatory pathway associated with other types of aneurysms [46,47].

Direct inhibition of NF-κB signaling with small molecules may prevent IA formation and progression. Dexamethasone is a synthetic corticosteroid drug that can suppress inflammatory signaling by preventing NF-κB’s translocation to the nucleus [48]. In the context of aneurysmal inflammatory pathways, it has shown benefits in non-cerebral models, such as murine thoracic aortic aneurysm and dissection (TAAD) [49]. Daily dexamethasone treatment significantly reduced inflammatory cell inflammation and ECM degradation in a β-aminopropionitrile monofumarate-induced TAAD mouse model [49]. Systemic corticosteroids may be less effective in IA because of their small size, but a corticosteroid-delivering coil using methods like Hoh et al.’s may provide comparable results [39]. Another administration method not investigated in the literature of corticosteroids for IAs is direct intra-arterial injection at the site of an aneurysm during endovascular treatment. Overall, IA pathogenesis is rich with targetable inflammatory molecules. IA therapies are administered systemically or with coated coils. Figure 1 provides a graphical summary of molecular targets for ECM remodeling.

## 3. VSMC Phenotypic Shift and ECM Remodeling

The aneurysm tissue not only distends in response to stress, but it also continues to thicken as the IA progresses [16]. As it grows, the flow within the IA becomes more turbulent, decreasing the shear stress and reducing endothelial cell regulation of vascular tone while simultaneously promoting continued remodeling through increased inflammation [50]. Increased inflammation within the IA continues to increase the levels of TNF-α, which eventually degenerates VSMCs and leads to their apoptosis [51]. VSMC apoptosis and inflammation recruit more inflammatory cells with the release of molecules such as monocyte MCP-1, IL-1, and IL-8 [38,51,52]. Wall remodeling is a component of IA formation, driven by molecules targetable for IA treatment.

MMPs secreted in IA walls contribute to ECM remodeling, causing weakness, which propagates further IA distension and rupture. In response to shear stress and inflammatory molecules, VSMCs undergo a phenotypic shift from contractile to synthetic, secreting MMP-2 and MMP-9 [53]. Molecular markers for synthetic VSMCs include S100A4, CD68, and the accumulation of lipids [54]. Increased MMP concentration in IAs correlates with decreased smooth muscle actin expression and increased NF-κB and MCP-1 in a rabbit model, illustrating remodeling of IA tunica media to a less dynamic state [53]. Interestingly, VSMCs seem to be the culprit of IA remodeling in this model, as macrophage depletion did not suppress IA development [53]. ECM remodeling weakens the vessel wall’s media layers and internal elastic lamina, thus leading to the advancement of the aneurysm [55,56]. Direct inhibition of MMP-2 and MMP-9 with a selective inhibitor did not show improvements in the aneurysm rupture rate in a murine model, implying MMPs do not act alone in IA rupture pathogenesis [57]. Based on this evidence, MMPs play a greater role in early IA formation but may be less viable molecular targets at a clinically relevant stage of IA development [55,57].

Endogenous tissue inhibitors of metalloproteinases (TIMPs) may protect from IA ECM remodeling by preventing the action of MMP-2 and MMP-9. VSMCs release TIMP-1 and TIMP-2 in the tunica media, within 15 days of IA induction in a rat model [58]. However, as the disease progresses, MMP expression outpaces TIMP expression [58]. The loss of parallel TIMP activity may be the cause of continued IA growth and rupture. TIMP knockout rats had significantly less ß-actin, marking less contractile VSMCs to respond to vascular stress, compared to wild-type mice [58]. Although IA incidence did not change in TIMP knockout mice, IA progression significantly increased [58,59]. Delivery of mRNA to increase the TIMP-1 ratio improves ECM integrity in animal models of abdominal aortic aneurysm; however, no literature exists on TIMP supplementation in IAs [60]. The TIMP/MMP ratio is a defining feature of IA progression, causing the delivery of molecules that increase the TIMP levels, a valuable IA therapeutic target.

Transforming growth factor-ß (TGF-ß) is another ECM-remodeling molecule implicated in IA progression. Human saccular IAs express significantly increased TGF-ß-R3 compared to healthy tissue, making TGF-ß a target for molecular IA therapy [60]. Specific literature on TGF-ß’s mechanistic role in IAs is unavailable. Further research should explore the role of TGF ligands and receptors with local inhibitors or knockout animal models to establish TGF-ß’s role in IA therapeutics. Delivery of therapies targeting ECM remodeling via coated coils would be ideal, as the systemic effects of reduced ECM remodeling activity could be detrimental to wound healing [61]. Another challenge will be delivery across the endothelial layer into the IA tunica media and adventitia, where ECM remodeling is taking place. Overall, intervening in IA progression via ECM remodeling is promising, but underexplored. Figure 2 provides a graphical summary of molecular targets for ECM remodeling.

## 4. Endothelial Dysfunction, Oxidative Stress, and Cell Death

Damage from wall shear stress and local inflammation disrupts endothelial nitric oxide synthase (eNOS), the primary regulator of vascular homeostasis, leading to IA formation [62]. Nitric oxide (NO) is a potent vasodilator and reactive oxygen species (ROS) generated by nitric oxide synthase (NOS) enzymes [63]. Increased stress on the lumen wall typically upregulates eNOS activity, but hemodynamic stress is low in IAs, which may lead to VSMC vasoconstriction and pro-inflammatory activity [64]. Polymorphisms in the eNOS gene rs1799983 change the risk of developing IAs, meaning eNOS may be an actionable target for IA therapy [65]. However, NOS inhibition with the small molecule L-arginine derivative, N(G)-nitro-L-arginine-methyl ester, reduces IA formation in a rabbit-induced IA model, counter to the logic that eNOS downregulation contributes to IA formation [66]. Endothelial dysfunction is a prelude to IA formation, facilitated by loss of eNOS, but other NOS species may be of greater importance in IA formation [67,68].

NO plays a greater role as a damaging, inflammatory ROS than a protective vasodilator in IA formation. Pathologically high NO levels, greater than VSMCs, can occur in IAs because their pro-inflammatory milieu induces the intrinsic NO synthase (iNOS) of infiltrating immune cells to form NO as a ROS for host defense [69]. Extremely elevated levels of NO downregulate eNOS via negative feedback, and NO works with other ROS from inflammatory cells to disrupt endothelial permeability and recruit more inflammatory cells via TNF-α and IL-1 release [69,70,71]. Non-selective NOS inhibitors reduce IA formation in animal models because of NO’s inflammatory role [66]. Interestingly, iNOS knockout mice have no increased rate of aneurysm formation, but when aneurysms do form, they have a greater size [72]. Endothelial dysfunction via reduced eNOS may play a role in IA formation, but NO’s role is primarily inflammatory once an IA forms, making iNOS and other generators of ROS more important molecular targets for IA therapy than eNOS.

One of the mechanisms that can result in damage in IA formation is through the production of ROS-generated inflammatory pathways. Aneurysmal blood flow is turbulent, specifically at points of bifurcation [73]. Turbulent flow increases reactive oxygen species production, which contributes to vascular wall injury and increases endothelial damage [38,72,74]. Increased ROS can oxidize lipids, damage mitochondrial DNA, and inactivate proteases [75]. Increased inflammation and impaired nitric oxide bioavailability result in VSMC apoptosis and disruption of the tunica media [76]. Additionally, ROS can directly activate transcription factors such as NF-κB, which further increase inflammatory markers [77,78]. They also disrupt NO synthesis, leading to endothelial dysfunction. Because ROS are key to IA formation, molecular treatments for IAs could target ROS or the molecules.

The NADPH pathway is key to ROS generation, but no investigations of NADPH in IAs exist in the literature [79]. In response to mechanical stress, endothelial and smooth muscle dysfunctions lead to increased NADPH oxidase activity, specifically NADPH oxidase (NOX) isoforms, including NOX1 and NOX2, in aneurysmal walls [80]. Specific NOX inhibitors can dampen ROS production and inflammation [80]. NADPH oxidase inhibitors significantly reduce abdominal aortic aneurysm formation [80]. Isoform-specific inhibitors for NOX1 and NOX2 are promising therapeutic agents in reducing the burden of oxidative stress in the central nervous system [80,81,82]. Further investigation of the NADPH pathway and NOX enzymes in IAs could provide another molecular target for IA therapy.

Resveratrol is an antioxidant polyphenol that reduces IA formation and rupture rates in mouse models [83,84]. Resveratrol induces epigenetic upregulation of antioxidative genes, superoxide dismutase, catalase, and FoxOs [85]. In a murine, hypertension-induced IA model, an RA-containing diet significantly inhibited the IA size and wall thickness compared to untreated mice [84]. Investigation of aneurysm walls revealed significantly lower macrophage infiltration, and Western blots measured lower MMP-2, MMP-9, and NF-κB in resveratrol-treated groups [84]. A similar intervention with dietary resveratrol in an elastase injection model of IA revealed no significant difference in IA formation, but a decrease in IA rupture rate [83]. By scavenging ROS to reduce oxidative stress, resveratrol may reduce the inflammation characteristic of IA formation and rupture [83,84].

RNA gene therapies could suppress NOX2/NOX4 signaling involved in oxidative damage or degrade pro-apoptotic inflammatory cytokines. Differential expression of microRNAs (miRNAs) is related to the ECM NOX function in aneurysmal tissue, such as miR-21, which can serve as a specific target for aneurysmal miRNAs [86,87]. An additional study recently developed neutrophil-mimetic nanoparticles coated with metal–organic frameworks to deliver siRNA targeting NOX4, with the goal of significantly reducing the ROS and ischemic injury in mouse models, highlighting the promise of targeted RNA therapies in vascular oxidative stress [88]. Overall, for molecular scientists to develop IA treatments targeting endothelial dysfunction and ROS, they will need to perform more mechanistic research.

## 5. Future Directions

More research to determine the initiating event of IA formation can help develop preventative, instead of remedial, therapies. Patients with IAs are usually asymptomatic, which makes studying IA initiation difficult [89]. Potential IA biomarkers discovered in proteomics studies are promising, but larger-scale investigations must confirm their efficacy [90,91]. IA biomarker candidates, such as MMP-9, are non-specific, but new artificial intelligence tools could discover more complex patterns in biomarker arrays for IA risk prediction, a growing trend seen in other fields, such as oncology [92]. Less invasive screening methods able to predict patients at risk of IAs will allow for systemic IA deployment. Developing biomarker screening tools to detect patients at risk of IA formation will be key to creating efficacious molecular IA therapies [93].

Circulating miRNAs are another area of promising IA research. A litany of miRNA species are reported to increase in the peripheral blood of patients with IAs compared to healthy controls [94,95,96,97]. The impetus to test for these miRNAs comes from genome-wide association studies reporting risk loci for IA formation [98,99]. These circulating miRNAs likely reflect the epigenetic shifts and molecular mechanisms of IA formation, but further investigation with larger clinical trials is needed to determine the sensitivity and specificity of each miRNA [100].

Unfortunately, many of the inflammatory molecules and cells implicated in IA progression are nonspecific and fundamental to survival, making continuous depletion with small molecules, interfering with RNA, or gene knockout difficult. One solution to this problem is spatiotemporally controlled mutant mice using the Cre-*loxP* system [101]. In brief, mice bred with *loxP* sites, flanking genes encoding a protein of interest in a specific tissue, have genes selectively inducible or inhibited by tamoxifen administration-activated Cre recombinase [101]. Halting tamoxifen administration reverses the effects of Cre [101]. This allows mice to be their own control in induced IA models, with growth measured with and without gene knockout by tamoxifen. The importance of different chemotactic cytokines, MMPs, and TIMPs at various stages of IA development is comparable using this method. Cre-*loxP* models would provide more validity in IA research, as most investigations focus on less-specific therapeutic molecules instead of a mechanistic explanation of IA formation.

Intravascular administration of small molecules or gene therapies targeting molecules implicated in induction, progression, and rupture is the ideal therapy for IAs [46]. Current guidelines for unruptured IA treatment recommend no specific systemic pharmacologic therapies [3]. Some retrospective observational studies suggest that patients taking anti-inflammatory medications such as aspirin reduce IA progression [102,103]. Systemic therapies targeting IL-6, TNF-α, and neutrophils are applied in other pathologies with inflammatory etiologies with great success, but leave patients immunocompromised [104,105,106]. Future randomized controlled trials of anti-inflammatory therapies targeting molecules implicated in IA progression must confirm that anti-inflammatory molecules are beneficial for patients with unruptured IAs, especially with their possible adverse effects. MMPs or TIMPs are not therapeutic targets used in current medical practice, providing less precedence for their use as a systemic therapy. As MMPs and TIMPs are ubiquitous, systemic therapies will likely have off-target effects, such as fibrosis [107]. Drugs targeting inflammation, MMPs, or ROS to reduce IA formation could be administered intra-arterially to reduce off-target effects of intravascular therapy, as calcium-channel blockers are administered for cerebral vasospasm in subarachnoid hemorrhage [108,109]. Overall, intravascular therapies for IAs targeting molecules implicated in IA formation, growth, and rupture are promising but will require diligent investigation in clinical trials to optimize dosage, aim at specific targets, and reduce adverse effects.

Designing systemic molecular IA therapies is difficult because IAs are a small compartment to target, even with direct intra-arterial administration. Molecular therapies should focus on deployment during endovascular procedures because coiling or stenting is the standard of care for IAs [110]. Most preclinical studies use direct, one-time delivery of therapy to IAs, which have a short duration of action. Gene therapies targeting molecules implicated in IA formation have the potential to provide a more sustainable therapy that can also penetrate deeper into IA layers [111]. More researchers should explore coil-coating as a delivery method, as this is the most promising application for implementing molecular IA therapies into the current IA standard of care [39,112]. Overall, future directions of molecular IA research include investigation of IA formation for screening biomarker development; more specific, temporal gene knockout studies to compare the importance of different molecules in IA progression; and exploration of pertinent routes of administration.

## 6. Conclusions

IAs are common and have a high mortality rate when they rupture [113]. Inflammation is a key mediator of IA formation. Chemotactic proteins such as MCP-1 and IL-8 facilitate immune cell infiltration into the endovascular wall, exacerbating an inflammatory cascade initiated by hemodynamic shear stress. The overall result is ECM remodeling to create a vascular wall unable to respond appropriately to stress. Temporally increasing the MMP-to-TIMP ratio is an important temporal pattern that molecular therapies could modulate. Another vital component of IA pathogenesis is endothelial dysfunction of reduced eNOS and increased synthesis of NO by iNOS, leading to vasoconstriction and formation of reactive oxygen species, further increasing inflammation. Future investigation into molecules associated with IA initiation, progression, and rupture aids the creation of tools to screen patients at risk of IA and molecular adjuvant therapies deployed during endovascular IA treatment.

## Figures and Tables

**Figure 1 ijms-26-10053-f001:**
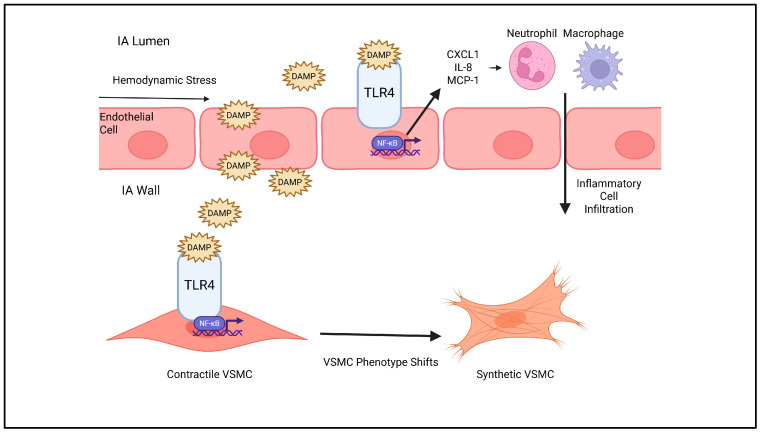
Inflammatory molecular pathways in intracranial aneurysm. Hemodynamic shear stress generates endothelial cell damage that generates damage-associated molecular patterns (DAMPs). DAMPs bind to TLR4 on endothelial cells and vascular smooth muscle cells (VSMCs), activating nuclear factor κB (NF-κB). Endothelial NF-κB activation leads to the secretion of inflammatory cell chemotaxis molecules, chemokine receptor ligand 1 (CXCL1), interleukin-8 (IL-8), and monocyte chemoattractant protein-1 (MCP-1), leading to neutrophil and macrophage infiltration in intracranial aneurysm cell walls. VSMC NF-κB activation shifts VSMC phenotype to remodel intracranial aneurysm cell walls.

**Figure 2 ijms-26-10053-f002:**
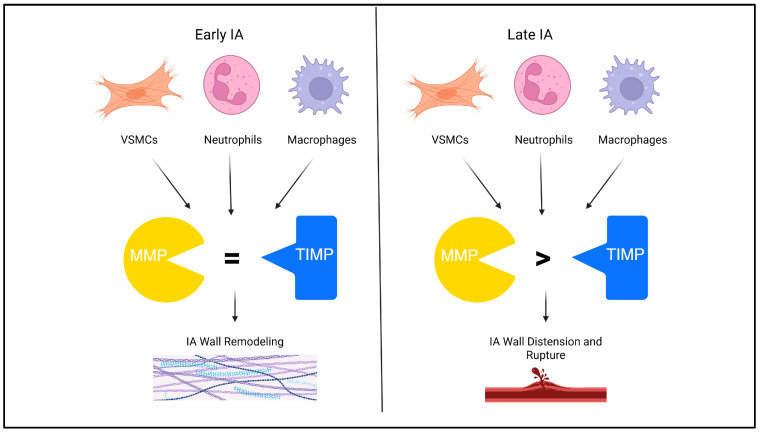
Extracellular matrix remodeling in intracranial aneurysm (IA) begins early, with matrix metalloproteinases (MMPs) balanced by tissue inhibitors of MMPs (TIMPs). In late IAs, the MMP-to-TIMP ratio increases, leading to wall distension and rupture.

## Data Availability

No new data were created or analyzed in this study. Data sharing is not applicable to this article.

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
