# Peer review of "Molecular Targets for Intracranial Aneurysm Treatment"

_ijms, 2025, doi:10.3390/ijms262010053_

Round 1
Reviewer 1 Report
Comments and Suggestions for Authors
The review article titled “Molecular Targets for Intracranial Aneurysm Treatment” is well written and an important study describes the role of inflammatory molecules in the development of IAs and targeting their expression for the treatment.
The authors describe possible signaling molecules involved in the development of IAs.
It would be advisable to incorporate antioxidant effects into the development of IAs, as oxidative stress plays an important role in the progression of IAs.
Author Response
Comment 1: It would be advisable to incorporate antioxidant effects into the development of IAs, as oxidative stress plays an important role in the progression of IAs.
Response 1: Thank you for your feedback to improve the manuscript. We added a paragraph on antioxidant modulation of IA formation with resveratrol to highlight targeting oxidative stress in molecular IA treatments in lines 279-288.
Reviewer 2 Report
Comments and Suggestions for Authors
In the present review, Hutchinson et al. discussed about Molecular Targets for Intracranial Aneurysm Treatment. The review is very interesting and well structured. However, it should be improved making some minor changes.
- In the abstract please add the acronym IA near Intracranial Aneurysm (first lane).
- In the paragraph “ECM remodeling” it is necessary to better clarify the specific phenotypic changes of VSMCs (add a line about the switch from contractile to synthetic phenotype). Please add the VSMCs also in the title of the paragraph together with ECM remodeling.
- The authors could also add a short paragraph on epigenetic factors (i.e. miRNAs) involved in IAs reported in the literature (there are several studies, see https://doi.org/10.1161/JAHA.114.000972).
Author Response
Comment 1: In the abstract please add the acronym IA near Intracranial Aneurysm (first lane).
Response 1: Thank you for your feedback. The change was addressed in line 10.
Comment 2: In the paragraph “ECM remodeling” it is necessary to better clarify the specific phenotypic changes of VSMCs (add a line about the switch from contractile to synthetic phenotype). Please add the VSMCs also in the title of the paragraph together with ECM remodeling.
Response 2: Thank you for this feedback. We added a better description of VSMC phenotypic shift from contractile to synthetic in lines 186-189. We also added changed the title of section 3 to "VSMC Phenotypic Shift and ECM Remodeling" in line 174.
Comment 3: The authors could also add a short paragraph on epigenetic factors (i.e. miRNAs) involved in IAs reported in the literature (there are several studies, see https://doi.org/10.1161/JAHA.114.000972).
Response 3: Thank you for your suggestion to strengthen the article. We added a paragraph on circulating miRNAs as a reflection of phenotypic shifts in lines 313-319.
Reviewer 3 Report
Comments and Suggestions for Authors Manuscript ID ijms-3892144Intracranial aneurysms (IA) develop as a result of complex processes such as hemodynamic stress, inflammation, and ECM degradation. Research on molecular therapeutic targets focuses on inflammatory pathways, matrix metalloproteinases, microRNAs, and novel omics-based targets. Although no drugs have been approved so far, targeted therapies such as inhibitors, statins, and gene therapies are being developed. This review article clearly and concisely presents both established and recent findings on IA pathogenesis and treatment. The authors also point out areas that warrant further research to improve diagnosis and therapy. The paper is a valuable resource for both clinicians and basic scientists whose research is translated into clinical practice. I recommend this manuscript for publication in the International Journal of Molecular Sciences.
Author Response
Thank you for your feedback on the manuscript.
Round 2
Reviewer 1 Report
Comments and Suggestions for Authors
The manuscript looks excellent and contains the required sections relevant to the study. I recommend that this manuscript be published without further revision.